# Variation in appropriate diabetes care and treatment targets in urban and rural areas in England: an observational study of the 'rule of halves'

Thomas Mason ![ID] , William Whittaker ![ID] , Jo C Dumville, Peter Bower ![ID]

Faculty of Biology, Medicine and Health, The University of Manchester, Manchester, UK

**Correspondence to**
Dr Thomas Mason;
thomas.mason@manchester.ac.uk

## ABSTRACT

**Objectives**  To estimate the 'rule of halves' for diabetes care for urban and rural areas in England using several data sources covering the period 2015–2017; and to examine the extent to which any differences in urban and rural settings are explained by population characteristics and the workforce supply of primary care providers (general practices).

**Design**  A retrospective observational study.

**Setting**  Populations resident in predominantly urban and rural areas in England (2015–2017).

**Participants**  N=33 336 respondents to the UK Household Longitudinal Survey in urban and rural settings in England; N=4913 general practices in England reporting to the National Diabetes Audit and providing workforce data to NHS Digital.

**Outcomes**  Diabetes prevalence; administrative records of diagnoses of diabetes; provision of (all eight) recommended diabetes care processes; diabetes treatment targets.

**Results**  Diabetes prevalence was higher in urban areas in England (7.80% (95% CI 7.30% to 8.31%)) relative to rural areas (7.24% (95% CI 6.32% to 8.16%)). For practices in urban areas, relatively fewer cases of diabetes were recorded in administrative medical records (69.55% vs 71.86%), and a smaller percentage of those registered received the appropriate care (45.85% vs 49.32%). Among estimated prevalent cases of diabetes, urban areas have a 24.84% achieving these targets compared with 25.16% in rural areas. However, adjusted analyses showed that the performance of practices in urban areas in providing appropriate care quality was not significantly different from practices in rural areas.

**Conclusions**  The 'rule of halves' is not an accurate description of the actual pattern across the diabetes care pathway in England. More than half of the estimated urban and rural diabetes population are registered with clinical practices and have access to treatment. However, less than half of those registered for treatment have achieved treatment targets. Appropriate care quality was associated with a greater proportion of patients with diabetes achieving treatment targets.

## INTRODUCTION

There are 55.9 million people in the UK residing in urban areas, 83.6% of the population.[1] Since 1960, the urban population of the UK has grown by 14.8 million and the rural population has decreased by around 370 000.[1] These national trends reflect wider global changes.[2] Demographic shifts towards urban areas have implications that include changes in residents' occupational profiles and health behaviours, and for non-communicable chronic illnesses such as diabetes.[3 4]

Previous evidence has shown that more than 70% of people with diabetes live in urban areas.[5] Conventional risk-factors such as body mass index, age, diet, family diabetes history and education do not fully explain differences in the diabetes risk of different economic groups of urban-residing individuals.[6] The Cities Changing Diabetes (CCD) programme was established in 2014 to generate further understanding of the burden of diabetes and its social and cultural determinants across a range of cities globally.[4]

One component of CCD is to describe the pattern of diabetes in terms of: prevalence; diagnosis; receipt of quality care; achievement of treatment targets and diabetes health outcomes. The 'rule of halves' (ROH) derives from the observation that numbers have been

### Strengths and limitations of this study

► We examined the extent to which differences in performance across the components of the diabetes care pathway were explained by population characteristics and supply factors using adjusted regression analyses.
► We estimated urban and rural prevalence of diabetes using nationally representative survey data.
► Practices returning data to the National Diabetes Audit (NDA) may be a selected sample and this could vary between the urban and rural practices.
► Patients can opt out from the NDA and may represent a selected sample.

observed to approximately halve at each level from prevalence to outcomes,[7] suggesting gaps in appropriate identification, management and treatment of diabetes along the care pathway.

In practice, the ROH represents a broad 'rule of thumb':[8] 'approximately half of most common chronic disorders are undetected, that half of those detected are not treated, and that half of those treated are not controlled'. While the 'halving' was found to apply for the British NHS in the 1980s for chronic illnesses including diabetes, hypertension and asthma,[8] more recent studies suggest that the ROH varies between countries for diabetes. Evidence from low-income and middle-income countries demonstrates this divergence. For example, estimates from India suggests a rule of 'two-thirds' (ie, a better performance than halving across levels),[9] whereas evidence from Peru comparing ROH patterns in urban and rural settings suggests considerably worse performance than halving on both care quality and treatment targets.[10] Applications of the ROH from high-income settings are equally inconsistent: evidence from Denmark outlined performance far in excess of 'halving'—in particular for diagnosing diabetes and providing treatment,[11] whereas evidence from Australia indicated that the ROH does in fact generally apply for diabetes care and management.[12]

Recent studies have updated historical evidence on the detailed pattern of the ROH for other chronic illnesses such as hypertension and osteoarthritis in England.[13–15] However, there are no recent efforts to update historical evidence on the ROH for England despite innovations in diabetes care and prevention over recent decades.[16–18] Additionally, despite the urban focus within the CCD programme and its corresponding research aims, the degree to which the ROH varies between urban and rural settings is not well established for England.

In some instances, studies estimating the ROH have examined how performance across each level varies according to conventional clinical risk factors such age, sex and comorbidity in addition to other population characteristics such as socioeconomic factors (eg, employment status).[10 11] Yet, there is limited evidence as to how supply factors (such as available resources in primary care)—which can confound socio-economic influences[19]—impact on performance across levels of the ROH.

In this study, we make three contributions. First, we update estimates on the pattern of the ROH for diabetes in England. Second, we quantify the extent to which the ROH and its subdomains represent an urban phenomenon in England (if at all) by comparing the ROH for urban and rural England. Finally, we examine the extent to which any urban/rural differences in the provision of appropriate care quality and achievement of diabetes care treatment targets can be explained by area-level differences in supply factors in addition to practice population characteristics.

## DATA
The ROH requires data on: diabetes prevalence rates and the estimated size of the population; the number of individuals with a recorded diagnosis of diabetes in administrative records; the number of individuals receiving appropriate diabetes care quality; and the number of individuals achieving appropriate diabetes treatment targets.

### Diabetes prevalence
We obtained estimated prevalence of diabetes using individual-level data at wave 7 of the UK Household Longitudinal Survey (UKHLS) (covering 2015–2017).[20] The survey started in 2009, initially comprising 40 000 households across the UK. The survey captures detailed information on mental and physical health in addition to a range of other topics including on urbanity/rurality of respondents' households. It contains large sample of individuals and its sampling methodology allows for examination of a nationally representative group of individuals. We used data for England only to derive an estimate of the mean annual prevalence of diabetes in England for the period 2015–2017.

### Total population estimates and rural/urban locations
We used estimates of population size at Clinical Commissioning Group (CCG) level by single year of age and sex for England in mid-2016 from the Office for National Statistics (ONS).[21] CCGs are responsible for the commissioning of health services for a defined regional area in England. We combined CCG population size data with ONS geographic data to capture urban and rural spread.[22] Additional detail on these data is provided in the online supplemental appendix.

### Diabetes registrations, care quality and treatment target achievement
We obtained practice-level data for England in 2016–2017 from the National Diabetes Audit (NDA).[23] NDA measures the effectiveness of diabetes care against National Institute for Health and Care Excellence (NICE) Clinical Guidelines and Quality Standards in England and Wales.[24] These data contain information on the number of individuals: with a diabetes diagnosis recorded in administrative records; receiving appropriate care quality and achieving appropriate treatment targets.

Care quality and treatment target achievement are captured in the NDA via indicators of eight care processes (which should be provided in line with clinical guidelines), and indicators of three treatment targets (that are appropriate for people with diabetes) (table 1). There are then two summary indicators of the numbers of patients achieving all eight care processes and three treatment targets respectively. Additional details are provided in the online supplemental appendix.

### Supply factors and practice population characteristics
Supply factors and practice population characteristics were measured using general practice list and workforce data for mid-2016–2017.[25] These data record numbers

**Table 1** Appropriate care processes and treatment targets recorded in the NDA

| NICE recommended annual care processes | |
| --- | --- |
| **Process** | **Detail** |
| HbA1c | Blood test for glucose control |
| Blood pressure | Measurement for cardiovascular risk |
| Serum cholesterol | Blood test for cardiovascular risk |
| Serum creatinine | Blood test for kidney function |
| Urine albumin/creatinine ratio | Urine test for early kidney disease |
| Foot risk surveillance | Foot examination for foot ulcer risk |
| Body mass index | Measurement for diabetes management |
| Smoking History | Question for cardiovascular risk |
| **NICE recommended treatment targets** | |
| **Target** | **Rationale** |
| HbA1c<58 mmol/mol | Target HbA1c reduces the risk of all diabetic complications |
| Blood pressure <140/80* | Target blood pressure reduces the risk of cardiovascular complications and reduces the progression of eye disease and kidney disease |
| Cholesterol <5 mmol/L | Target cholesterol reduces the risk of cardiovascular complications |

*See Refs. 23 and 24. NICE recommendations have been revised since 2016–2017 to recommend blood pressure targets at 140/90; and guidelines for best practice/care quality is continuously updated.[32]
NDA, National Diabetes Audit; NICE, National Institute for Health and Care Excellence.

and details of general practitioners (GPs) in England along with information on their practices, staff and patients. Specifically, the data record general practices' list size and the age/sex composition of their registered populations. These demographics are important determinants of need. The data also provide information on the number of full-time equivalent (FTE) GPs, nurses and administrative staff. These measures allow for the capture of supply factors critical to meeting the need for primary care arising in registered populations.

## METHODS
### Populating the levels of the ROH for urban and rural CCGs
#### Estimating mean diabetes prevalence for England (2015–2017)
We combined data from wave 7 (2015–2017) of the UKHLS on participants' self-reports of health conditions with information from a nurse visit assessment in waves 2 and 3 (2010–2012) including measurement of glycated haemoglobin (HbA1c). We used this information to create an indicator of whether a participants had diabetes

or not at wave 7 to align with the time period covered by the other data sources in this study.

We then used the UKHLS measure of urbanity/rurality of respondents' area of residence to stratify individuals into two groups (urban or rural). This measure classifies respondents addresses as falling into either an urban or rural area based on ONS Rural and Urban Classification of Output Areas.[26] In practice, this means that respondents are classified as urban if their address falls within urban settlements with a population of 10 000 or more. We estimated prevalence of diabetes in these two groups at wave 7 (2015–2017) including corresponding CIs, applying appropriate longitudinal sample weights.[27]

### Assessing diabetes registrations, care quality and treatment target achievement
We aggregated data from the NDA at practice-level to CCG-level. These data provide the number of: diabetes registrations; patients receiving all eight care processes; and patients achieving all three treatment targets. Data on care quality from the NDA assume that patients without data for a measurement have not met the criteria for achieving an individual indicator.[24] We excluded (N=38) CCGs classed as 'urban with significant rural' to allow for comparison between more distinctly urban and rural CCGs (however, these CCGs are included in online supplemental analyses). This provided totals of each for the (N=121) predominantly urban and (N=32) predominantly rural CCGs in England. These are summarised in descriptive tables and figures in the online supplemental material.

### The ROH at national-level in urban and rural areas
To estimate total population prevalence in urban and rural settings in England, we aggregated mid-2016 population estimates by age and sex to total figures for (N=121) predominantly urban and (N=32) predominantly rural CCGs in England. However, we applied an adjustment to these population estimates in order to produce a measure of total population prevalence in mid-2016 that would be comparable with the administrative data on diabetes registrations, care quality and treatment target achievement. Practice participation in the NDA is on average 95.3 per cent, and this varies slightly across individual CCGs. We therefore weighted estimated CCG population size by the corresponding figure for practice participation in the NDA. This allowed for consistent comparison of performance between total prevalence and the other levels of the ROH in predominantly urban and predominantly rural CCGs.

### Adjusted regression analyses of variation in care quality provision and treatment target achievement
#### Assessing variation in appropriate care quality between urban and rural CCGs and the influence of population features and supply factors
We used regression analyses to examine whether the percentage of registered people with diabetes receiving all eight care processes varied between general practices in predominantly urban and predominantly rural

CCGs, and to what extent any difference might have been explained by variation in urban and rural practices' population characteristics and workforce. We include additional variables iteratively to clearly show the impact of including additional controls on the main estimate of interest (the binary indicator for practice membership of a predominantly urban CCG).

First, we included a binary indicator as to whether a practice was located in a predominantly urban CCG. We then included the characteristics of practices' diabetic and overall populations (binary indicators for the size of the registered diabetic population (under 250; 500–749; 750 and over (with 250–500 as the reference)]; % of diabetics in age intervals (% under 40; % 40–64; % 65–79); % male; % of diabetics from most deprived 40% neighbourhoods based on Index of Multiple Deprivation score; total practice list size (divided by 1000) and the number of patients by sex in each of the following age groups (under 44; 45–64; 65–84; 85 and over)). Finally, we included the number of FTE GPs, nurses and administrative staff per 1000 patients.

### Assessing variation in treatment target achievement between urban and rural CCGs and the influence of population features and supply factors

We then repeated the above analyses instead using the percentage of registered patients with diabetes achieving all three treatment targets as the outcome variable. First, we included the indicator for predominantly urban CCGs, and then included the percentage of registered patients with diabetes receiving all eight care processes to control for this component of the ROH. We then included the same measures of practice population characteristics and workforce supply detailed above.

All regression analyses were performed using weighted ordinary least squares at practice-level. The numbers of registered diabetics at practice-level were used as weights, and models were clustered by general practice. We excluded n=42 observations on N=19 practices from the NDA data for which we had no data on workforce/list characteristics from the regression analyses.

### Patient and public involvement

Patients and the public were not involved in this study, but results were fed back via community engagement work undertaken as part of a wider research project which includes qualitative work informed by patient engagement. This study uses retrospective, observational data (most of which are at an aggregated level) and did not require ethical approval.

## RESULTS

A range of supplementary findings are presented and explained in the supplementary material (online supplemental tables S1–S8; online supplemental figure 1).

### The ROH at national-level

The corresponding aggregation of the CCG levels of the ROH to national-level for urban and rural areas are set out in table 2 and figure 1.

Aggregation of the urban and rural picture to national-level outlines the pattern of the ROH in each setting. In predominantly urban areas, 69.55% of total estimated prevalence is registered with a diagnosis in administrative records (1.78 million of 2.56 million) (table 2). This compares with higher overall rates of registration in rural areas (71.86% (450 850 of 627 431)).

In total, practices in urban areas provided appropriate care for 45.85% of patients with a record of diabetes (813 000 of 1.78 million) (table 2). Overall, practices in rural areas provided 49.32% of patients with a diabetes record with appropriate care (222 345 of 450 850). The relatively lower rate of providing appropriate care in urban areas compounds the lower rate of registration in the prevalent population: 31.89% of estimated prevalence receives appropriate care for urban areas, compared with 35.33% in rural areas (in relative terms, a difference of 9.74%).

**Table 2** National levels of the ROH by urban/rural status

|  | Predominantly urban CCGs (n=121) | | | Predominantly rural CCGs (n=32) | | |
|  | Estimated total population*=32 877 630 | | | Estimated total population*=8 699 248 | | |
|  | Total | % of previous level† | % of prevalence | Total | % of previous level† | % of prevalence |
|---|---|---|---|---|---|---|
| Diabetes prevalence | 2 566 014 | 7.80 | 100.00 | 627 431 | 7.24 | 100.00 |
| Registrations | 1 784 715 | 69.55 | 69.55 | 450 850 | 71.86 | 71.86 |
| Appropriate care quality | 818 300 | 45.85 | 31.89 | 222 345 | 49.32 | 35.44 |
| Achieve treatment targets | 637 325 | 77.88 | 24.84 | 157 880 | 71.01 | 25.16 |

*Total population adjusted for practice participation in NDA; diabetes prevalence estimated using UKHLS data for Wave 7 (2015–2017).
†Previous level refers to row above (as denominator), prevalence refers to as % of estimated total population.
CCG, Clinical Commissioning Group; NDA, National Diabetes Audit; ROH, 'rule of halves'; UKHLS, UK Household Longitudinal Survey.

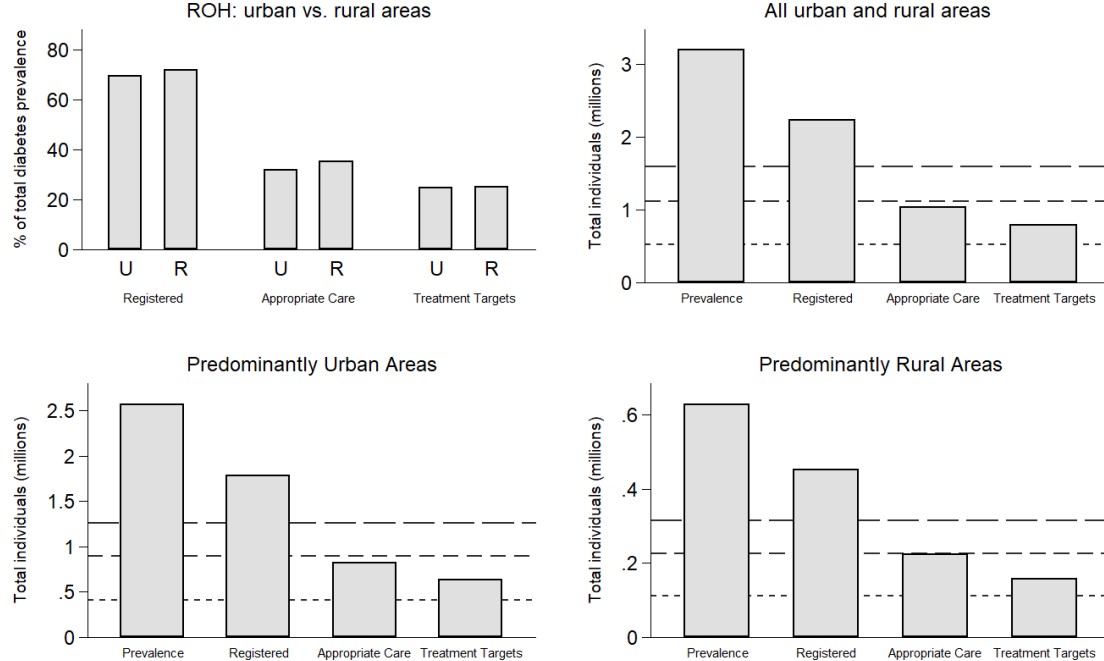

**Figure 1** ROH comparison across levels for urban and rural areas in England (2016–2017). ROH, 'rule of halves'.

Finally, practices in urban areas had patients with a record of diabetes achieve their treatment targets at an overall rate of 77.88% among those receiving the appropriate care (637 325 of 818 300). This compares with an equivalent rate of 71.01% in rural areas (157 880 of 222 345). However, this higher overall performance for urban areas among those receiving appropriate care reflects both lower rates of registration and provision of appropriate care in urban areas. Among estimated prevalence, urban areas have 24.84% achieving these targets (637 325 of 2.56 million); compared with 25.16% in rural areas (157 880 of 627 431).

### Adjusted analyses of variation in appropriate care quality between urban and rural CCGs

Predominantly urban CCGs had 3.37% fewer patients receiving all eight care processes relative to predominantly rural CCGs (p<0.001) (table 3). However, this difference fell to 1.908% (p<0.05) after adjusting for the effect of the characteristics of the practice overall and the diabetic population (table 3). Finally, there was no statistically significant difference in the performance of predominantly urban and rural practices after further adjusting for indicators of practice workforce supply.

Practices with fewer than 250 patients registered with a record of diabetes had 3.58% (p<0.001) fewer patients achieving all eight care processes compared with practices with between 250 and 499 patients with a diabetes record (table 3). Practices with larger list sizes had fewer patients receiving all eight care processes: 0.54% for each additional 1000 patients (p<0.001). Each additional GP per 1000 patients was associated with a 0.53% increase in registered patients with diabetes achieving all eight care processes (table 3).

### Analyses of variation in treatment target achievement between urban and rural CCGs

Predominantly urban CCGs had 0.755% more patients recorded as achieving all three treatment targets relative to predominantly rural CCGs (p<0.001) (table 4).

This difference increased to: 1.044% (p<0.001) after adjusting for the effect of the percentage of patients receiving all eight care processes; 1.402% (p<0.001) after adjusting for the characteristics of practices overall and diabetic populations and 1.309% (p<0.001) after including measure of the practice workforce supply (table 4).

Practices with smaller diabetic populations (fewer than 250 registered) had 1.67% more patients achieving all three treatment targets compared with practices with 250 to 499 patients with a diabetes record (p<0.001). For each additional 10% of patients with a record of diabetes being aged under 40, practices had 2.32% fewer patients achieving all three treatment targets (p<0.001); and for each additional 10% of registered patients with diabetes being aged 65–79, practices had 2.58% more fewer patients achieving all three treatment targets (p<0.001) (table 4).

### DISCUSSION
### Summary of key findings

For practices in predominantly urban areas in England, the ROH displayed a different pattern compared with rural areas for 2016–2017. Diabetes prevalence was higher in urban CCGs in England (7.80%) relative to rural areas (7.24%) despite the population being considerably younger. Relatively fewer cases of diabetes were recorded in administrative medical records (69.55% vs 71.86%),

**Table 3** Regression estimates—analyses of the % of patients receiving appropriate diabetes care quality in general practices

| | I | II | III |
|---|---|---|---|
| **Urban/rural status** | | | |
| Predominantly rural=1 | | *Reference group* | |
| Predominantly urban=1 | −3.370*** | −1.908* | −1.488 |
| **Practice diabetic population characteristics** | | | |
| Registered diabetics (<250)=1 | | −3.340*** | −3.576*** |
| Registered diabetics (250–499)=1 | | *Reference group* | |
| Registered diabetics (500–749)=1 | | 2.217** | 2.379** |
| Registered diabetics (750+)=1 | | 2.897* | 3.029* |
| % of diabetics aged under 40 | | 0.209 | 0.236* |
| % of diabetics aged 40–64 | | −0.071 | −0.069 |
| % of diabetics aged 65–79 | | 0.122 | 0.134 |
| % of diabetics=male | | 0.047 | 0.066 |
| % patients from most deprived 40% neighbourhoods | | 0.043** | 0.043** |
| % of patients from ethnic minorities | | −0.090*** | −0.088*** |
| % of patients of unknown ethnicity | | −0.063*** | −0.061*** |
| List size (1000 s) | | −0.065 | −0.545*** |
| **Practice labour supply** | | | |
| GPs per patient | | | 0.530*** |
| Nurses per patient | | | 0.255 |
| Admin staff per patient | | | 0.023 |
| Constant | 49.392*** | 48.814*** | 48.731*** |
| R-sq. | 0.005 | 0.042 | 0.049 |
| n | 4913 | 4913 | 4913 |

*P<0.05; **p<0.01; ***p<0.001; Model I=Urban indicator only; Model II=I+Practice population characteristics; Model III=II+Practice labour supply; Models II and III include indicators for the age structure of practice list (by sex); all models estimated at practice-level using Ordinary Least Squares (OLS) weighted by the number of registered diabetics; all models clustered by general practice.

and a smaller percentage of those registered received the appropriate care (45.85% vs 49.32%). However, practices in urban areas achieved a slightly higher percentage of those receiving appropriate care then achieving treatment targets (24.84% vs 25.16%). Adjusted analyses showed that the lower performance of practices in urban CCGs in providing appropriate care quality did not persist after adjusting for both the needs of the populations they served and the practice workforce with which they were tasked with meeting those needs.

### Explanation of notable findings

Examination of the aggregate performance indicators on provision of appropriate care by practices in urban and rural areas would superficially indicate the provision of a lower level of care quality by urban practices. Yet, the adjusted analyses in this study show that these differences do not persist given the populations served and supply factors. In fact, after adjustment: the performance of urban practices was not statistically different in terms of care quality and was better in terms of treatment target achievement.

Differences in the proportion of patients with a record of diabetes receiving the appropriate care could in theory reflect differences in attitudes and health literacy among urban and rural populations. Practices in urban settings typically serve younger, more deprived and ethnically diverse populations (online supplemental table S3). Previous studies have shown that education, age, gender and ethnicity have the strongest associations with health literacy in rural and urban areas; and that urbanity/rurality does not represent a specific determinant of health literacy.[28] Adjustment for the population characteristics of practices in urban areas substantially reduced the performance gap but did not fully account for the difference. This gap was no longer significant after further adjusting for supply factors. This may reflect inadequate capture of differences in the needs of diabetic populations served by urban and rural practices in the allocation of resources to practices.

Adjusted analyses confirmed that practices serving rural populations tend to have lower rates of treatment target achievement, even after adjusting for population and workforce characteristics. The data did not allow

**Table 4** Regression estimates—analyses of the % of (registered) patients with diabetes achieving diabetes treatment targets in general practices

| | I | II | III | IV |
|---|---|---|---|---|
| **Urban/rural status (reference=predominantly rural)** | | | | |
| Predominantly rural=1 | Reference group | | | |
| Predominantly urban=1 | 0.755** | 1.044*** | 1.402*** | 1.309*** |
| **Appropriate care quality** | | | | |
| % of patients receiving appropriate care | | 0.086*** | 0.087*** | 0.088*** |
| **Practice diabetic population characteristics** | | | | |
| Registered diabetics (<250)=1 | | | 1.638*** | 1.676*** |
| Registered diabetics (250–499)=1 | | | Reference group | |
| Registered diabetics (500–749)=1 | | | −0.593* | −0.614* |
| Registered diabetics (750+)=1 | | | −0.04 | −0.015 |
| % of diabetics aged under 40 | | | −0.231*** | −0.232*** |
| % of diabetics aged 40–64 | | | 0.052 | 0.053 |
| % of diabetics aged 65–79 | | | 0.260*** | 0.258*** |
| % of diabetics=male | | | −0.019 | −0.021 |
| % of patients from most deprived 40% neighbourhoods | | | −0.009 | −0.009 |
| % of patients from ethnic minorities | | | 0.030*** | 0.029*** |
| % of patients of unknown ethnicity | | | −0.025*** | −0.025*** |
| List size (1000 s) | | | −0.086* | −0.022 |
| **Practice labour supply** | | | | |
| GPs per patient | | | | −0.031 |
| Nurses per patient | | | | −0.069 |
| Admin staff per patient | | | | −0.011 |
| Constant | 35.005*** | 30.781*** | 34.512*** | 7.971 |
| R-sq. | 0.002 | 0.054 | 0.128 | 0.129 |
| n | 4913 | 4913 | 4913 | 4913 |

*P<0.05; **p<0.01; ***p<0.001; Model I=Urban indicator only; Model II=I+Care quality; Model III=II+Practice population characteristics; Model IV=III+Practice labour supply; Models III+IV include indicators for the age structure of practice list (by sex); all models estimated at practice-level using OLS weighted by the number of registered diabetics; all models clustered by general practice.

for exhaustive examination of the possible underlying causes of this. However, there are possible explanations which may warrant further investigation in future research. Patients registered with rurally located practices typically have further to travel to access primary care, and those located furthest away exhibit a lower propensity to use services.[29–31] Maintenance of appropriately controlled HbA1c levels in patients with diabetes typically requires prescription of metformin, the standard first line of therapy.[32] However, in patients who are inappropriately controlled by metformin, intensification of therapy is required (with additional prescription drugs). Future studies might investigate whether rurally residing patients with diabetes exhibit a lower propensity to seek out primary care when they are becoming more poorly controlled.

### Strengths/comparison with existing research

We updated estimates on the pattern of the ROH in urban and rural England by combining: nationally representative survey data for 2015–2017; demographic and geographic data for mid-2016 and administrative data on diabetes care and treatment targets with national coverage for 2016–2017.

We derived an independent estimate of diabetes prevalence in urban and rural areas for England using large, nationally representative survey data. Survey data are well suited to derive estimated disease prevalence for conditions such as diabetes as they minimise the risk of estimates being based on a potentially self-selecting group of individuals such as those in contact with health services.[18]

We compared the pattern of the ROH in urban and rural areas to understand the extent to which it may represent an urban phenomenon in England, and at which levels of the ROH the largest differences in performance exist. We then used adjusted regression analyses to examine the extent to which underlying determinants of need and practice workforce supply might explain differences in the performance of

practices in urban and rural areas at particular levels of the ROH.

## Limitations

We adjusted population estimates for NDA practice participation to allow for consistency across data sources, and therefore estimates at an aggregate level will likely understate the totals for each level of the ROH. Practices returning data to the NDA may be a selected sample and this could vary between the urban and rural CCGs practices. Additionally, patients can opt out from the NDA and may represent a selected sample biasing the NDA practice-level data.

We used a range of data sources to populate the ROH, and these corresponding time periods across data sources were imperfect—for example, we used mid-2016 population estimates to compare with 2016–2017 data from the NDA. This reflected best availability, but the findings should be interpreted in this context in light of the minor differences between urban and rural areas. Data on diabetes registrations, care processes and treatment targets were for the financial year 2016–2017. However, the guidelines around best practice care are updated continuously and the findings of this study should be interpreted on this basis. There are five pillars of the ROH, and we excluded the final pillar (diabetes-related health outcomes) from this study due to poor availability of high quality data.

## Implications for research and policy

The ROH is a heuristic and is useful in drawing attention to how gaps occur across the different aspects of the care pathway, but it is not an accurate description of the actual pattern in England. More than half of the estimated urban and rural diabetes population are registered with clinical practices and have access to treatment. However, less than half of those registered for treatment have achieved treatment targets. The aggregate differences between practices in urban and rural settings are less pronounced after reflecting the differences in the populations served and the workforce used to provide care.

While the differences between English rural and urban settings in terms of performance are relatively small, this does not mean that the causes are the same. Further qualitative research is needed to further understand why drop-offs occur across the levels of the ROH in urban settings. Future studies could also explore whether health literacy of patients with diabetes does in fact explain some of the variation between patients residing in urban and rural settings. This evidence could then be used to help inform development of evidence-based health policy and clinical practice for urban populations, for example, by improving health literacy.

**Contributors** PB, WW and JCD secured the funding. TM, WW, JCD and PB designed the analysis. TM performed the analysis. TM drafted the paper and PB, WW, and JCD critically revised it. The corresponding author attests that all listed authors

meet the relevant criteria and no other meeting these have been omitted. TM acts as the guarantor for this study.

**Funding** This research was supported by the National Institute for Health Research Applied Research Collaboration Greater Manchester. This research originates from work undertaken in the Cities Changing Diabetes (CCD) programme for Greater Manchester in partnership with and supported by Health Innovation Manchester and Novo Nordisk. CCD was launched in 2014 by the Steno Diabetes Center Copenhagen, University College London, and Novo Nordisk.

**Disclaimer** The views expressed in this publication are those of the author(s) and not necessarily those of the National Institute for Health Research or the Department of Health and Social Care.

**Competing interests** None declared.

**Patient consent for publication** Not applicable.

**Provenance and peer review** Not commissioned; externally peer reviewed.

**Data availability statement** Data may be obtained from a third party and are not publicly available. Data sharing statement: the population and practice-level data from this study are publicly available from the Office for National Statistics and NHS Digital, the data used to generate prevalence estimates from UK Household Longitudinal Survey can be applied for via the UK Data Service.

**ORCID iDs**
Thomas Mason http://orcid.org/0000-0003-3135-0364
William Whittaker http://orcid.org/0000-0003-2530-0360
Peter Bower http://orcid.org/0000-0001-9558-3349

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
