## [Reviewer comments · BMJ Open]

ARTICLE DETAILS

TITLE (PROVISIONAL)	Variation in appropriate diabetes care and treatment targets in urban and rural areas in England: an observational study of the 'rule of halves'
AUTHORS	Mason, Thomas; Whittaker, William; Dumville, Jo C.; Bower, Peter

VERSION 1 – REVIEW

REVIEWER	Bernabe-Ortiz, Antonio Universidad Peruana Cayetano Heredia, CRONICAS Centre of Excellence in Chronic Diseases
REVIEW RETURNED	23-Sep-2021

GENERAL COMMENTS	Overall, this is a very interesting paper using population-based data to estimate the ROH for diabetes care in UK. Valuable information is given in this manuscript; however, there are some things that need better presentation. Abstract: The abstract has not enough information compared to that given in the manuscript. Usually, authors do not have enough space for writing all the details in the abstract, but I feel that this abstract give not enough justice to the paper. For example, not clear why in the objectimes time framework is 2016-17, and in the setting is 2015-2017 (which is clear in the main document). Similarly, some outcomes are assumed to be understandable, but it is not the case for practices, or appropriate diabetes care quality, etc. Finally, some parts of the first sentence of the conclusions section are not relevant. Main manuscript: - Some acronyms should be defined before using them.- Blood pressure in Table 1 should be 140/90 according to updated NICE recommendations. Understandable if using previous NICE recommendations as data is from 2016-2017, but need to be clarified in the table.- A concern about the exclusion of 38 CCGs... maybe to show a comparison between the three groups as an online suplement to appropriately understand if the exclusion was valid or not.- Tables 3 and 4 needs be self-explicative. In some cases, it is not clear what the reference group is in the models.- Authors did not discuss why some indicators are better in rural compared to urban settings (registration and care quality), and despite of that, targets did not follow the same pattern.- Some emphasis should be put on the fact that small differences reported may not be as relevant as authors initially thought.
---

REVIEWER	Dickson, Lynnsay University of the Witwatersrand, Paediatrics
-----------------	--

GENERAL COMMENTS

Thank you for this report.
Your work is most appreciated.

Queries are as follows:

2. Abstract

Results:

The following comment is potentially misleading: "However practices in urban areas achieved a higher percentage of those receiving appropriate care than achieving treatment targets (77.88% vs. 71.01%)"

[line 46-48; page 3]

The following quotation from the results section of this article reflects a more accurate treatment outcome in this population: "among estimated prevalence urban areas have a 24.84% achieving these targets compared with 25.16% in rural areas".

[line 25-27; page 13]

Conclusion:

The following comment is potentially misleading: "Performance is considerably better than halving in terms of diagnoses in care records and achievement of treatment targets."

[line 3-6; page 4]

The rule of halves is "Approximately half of most common chronic disorders are undetected, that half of those detected are not treated, and that half of those treated are not controlled".

[Reference: Hart, JT. Rule of halves: implications of increasing diagnosis and reducing dropout for future workload and prescribing costs in primary care. British Journal of General Practice. 1992; 42:116-119]

Better than half of the estimated urban and rural diabetes population (69.55% vs 71.86%) are registered with clinical practices and have access to treatment. However, less than half of those registered for treatment have achieved treatment targets (35.71% vs 35.02%). [Supplementary table S2; page 26]

The adherence to/ or effectiveness of the local NICE guidelines is not a component of the rule of halves and is not the objective of the study. It is important to document that appropriate quality care/ adherence to these guidelines, when delivered to urban and rural patients (45.85% vs 49.32%), is associated with a greater likelihood of achieving treatment targets. (77.88% vs 71.01%).

4. Methods

The strategy for management of missing patient clinical data is not explicitly stated. Is this the definition of not achieving appropriate care?

5. Ethics

No reference is made to ethical considerations.

11. Discussion

With regard to your comments: "Differences in the proportion of patients with a record of diabetes receiving the appropriate care

	could in theory reflect differences in attitudes and health literacy amongst urban and rural populations" [line 3-7 page 17]; and "This evidence could then be used to help inform development of evidence based health policy and clinical practice for urban populations for example by improving health literacy." [lines 49-51; page 18]: Do you have access to data on patient diabetes health literacy, e.g. participation in the NICE DESMOND program (diabetes education for self-management for ongoing and newly diagnosed)? Regarding: "Adjustment for the population characteristics of practices in urban areas substantially reduced the performance gap but did not fully account for the difference" [line13-18; page 17]: Information available to the investigators included the mental health status of urban and rural populations. [line 4-6; page 7] Was this included in further analyses? For reasons stated previously, consider amending this misleading comment in the conclusion: Performance is considerably better than having in terms of diagnosis in key records and achievement of treatment targets. [line 37-39; page 18] 13. Funding The source of funding is not defined. [line 15-16; page 19]
--	--

REVIEWER	Li, Sheyu Department of Endocrinology and Metabolism, West China Hospital, Sichuan University
REVIEW RETURNED	05-Oct-2021

GENERAL COMMENTS	The study based on a series of surveys illustrates the quality of care for diabetes in England, UK, and indicates some but not strong inequity of health care. There are some concerns regarding the analysis.  1. The quality of care is confusing and Table 1 needs citation to support the rationales. To be noted, ideal care strategy changes over time. The citation should fit the time point of care, and the time lag may be considered for the knowledge dissemination. To be noted, NICE is updating itself continuously at different aspects of diabetes care. 2. The data resource and definition for diabetes registration is critical and needs more clarification. 3. The study differentiates only urban and rural areas in England. It might be more helpful to indicate the healthcare inequity across counties or shires of England. 4. I am confused about the meaningfulness of the "ROH", especially the term "half". The paper says little about half, but different numbers for different events. I do not think that removing the term "ROH" eliminates the significance of the study.
---

VERSION 1 – AUTHOR RESPONSE

Reviewer: 1

Dr. Antonio Bernabe-Ortiz, Universidad Peruana Cayetano Heredia, Universidad Científica del Sur

Comments to the Author:

Overall, this is a very interesting paper using population-based data to estimate the ROH for diabetes care in UK. Valuable information is given in this manuscript; however, there are some things that need better presentation.

Abstract:

The abstract has not enough information compared to that given in the manuscript. Usually, authors do not have enough space for writing all the details in the abstract, but I feel that this abstract give not enough justice to the paper. For example, not clear why in the objectives time framework is 2016-17, and in the setting is 2015-2017 (which is clear in the main document).

We have now revised the objectives to make clearer to the reader that we combine a range of data sources from the period 2015-17 for consistency with the 'setting' section of the abstract, and the main text document (page 3):

“Objectives

To estimate the 'rule of halves' for diabetes care for urban and rural areas in England using several data sources covering the period 2015-17; and to examine the extent to which any differences in urban and rural settings are explained by population characteristics and the workforce supply of primary care providers (general practices)”

Similarly, some outcomes are assumed to be understandable, but it is not the case for practices, or appropriate diabetes care quality, etc.

We have now defined general practices as primary care providers to clarify this to an international audience (see previous response)

We have also revised the definition of care quality to more clearly show that this is defined with respect to the provision of all eight diabetes care processes (page 3):

“Outcomes

Diabetes prevalence; administrative records of diagnoses of diabetes; provision of (all eight) recommended diabetes care processes; diabetes treatment targets”

Finally, some parts of the first sentence of the conclusions section are not relevant.

This has been revised to now remove the content around the ROH as a heuristic (page 4):

“Conclusions

The 'rule of halves' is not an accurate description of the actual pattern across the diabetes care pathway in England. More than half of the estimated urban and rural diabetes population are registered with clinical practices and have access to treatment. However, less than half of those registered for treatment have achieved treatment targets. Overall, performance across the levels of the rule of halves is lower for urban areas - but differences are explained by population characteristics and supply factors. Appropriate care quality was associated with a greater proportion of diabetic patients achieving treatment targets”

Main manuscript:

- Some acronyms should be defined before using them.

The manuscript has been carefully checked for acronyms to ensure they are properly defined, and revised where required.

- Blood pressure in Table 1 should be 140/90 according to updated NICE recommendations. Understandable if using previous NICE recommendations as data is from 2016-2017, but need to be clarified in the table.

Table 1 has been revised to include the following footnote (page 8):

“*note: NICE recommendations have been revised since 2016-17 to recommend blood pressure targets at 140/90”

- A concern about the exclusion of 38 CCGs... maybe to show a comparison between the three groups as an online supplement to appropriately understand if the exclusion was valid or not. We have repeated all descriptive and regression analyses (including corresponding versions of all tables and figures), but included the N=38 ‘urban with significant rural’ CCGs. The results from these analyses do not change the overall pattern of findings or corresponding interpretation, and the results have been included in the supplement to reassure the reader that the exclusion was valid. This material is presented in the results section of the supplementary material under the heading:

“Extension to include ‘urban with significant rural’ CCGs” (pages 28 to 34), and includes the following tables and figures:

- Supplementary Tables S4 - S8
- Supplementary Figure S1

This supplement is also referred to in the text in the methods section which refers to the exclusion of the 38 ‘urban with significant rural’ CCGs (page 9):

“We excluded (N=38) CCGs classed as “urban with significant rural” to allow for comparison between more distinctly urban and rural CCGs (however, these CCGs are included in supplementary analyses).”

- Tables 3 and 4 needs be self-explicative. In some cases, it is not clear what the reference group is in the models.

Rows have been added in to Tables 3 and 4 to clarify the reference groups in the analyses (there are two – predominantly rural practices, and practices with 250-499 registered diabetic patients in total). See Tables 3 and 4 on pages 14-16.

- Authors did not discuss why some indicators are better in rural compared to urban settings (registration and care quality), and despite of that, targets did not follow the same pattern.

We have now added in an additional paragraph discussing this result, and how future studies might look into whether differences in health care seeking behaviour of rurally residing diabetics leads to them being more poorly controlled (for example, if they were to be in possible need of having an intensification of therapy (pages 17 and 18):

“Adjusted analyses confirmed that practices serving rural populations tend to have lower rates of treatment target achievement, even after adjusting for population and workforce characteristics. The data did not allow for exhaustive examination of the possible underlying causes of this. However, there are possible explanations which may warrant further investigation in future research. Patients registered with rurally located practices typically have further to travel to access primary care, and

those located furthest away exhibit a lower propensity to use services (29–31). Maintenance of appropriately controlled HbA1c levels in patients with diabetes typically requires prescription of metformin, the standard first line of therapy (25). However, in patients who are inappropriately controlled by metformin, intensification of therapy is required (with additional prescription drugs). Future studies might investigate whether rurally residing diabetes patients exhibit a lower propensity to seek out primary care when they are becoming more poorly controlled.”

- Some emphasis should be put on the fact that small differences reported may not be as relevant as authors initially thought.

We have now added the following statement in the discussion (implications for research and policy, page 19):

“The aggregate differences between practices in urban and rural settings are less pronounced after reflecting the differences in the populations served and the workforce used to provide care.”

Reviewer: 2

Dr. Lynnsay Dickson, University of the Witwatersrand

Comments to the Author:

Thank you for this report.

Your work is most appreciated.

Queries are as follows:

2. Abstract

Results:

The following comment is potentially misleading: “However practices in urban areas achieved a higher percentage of those receiving appropriate care then achieving treatment targets (77.88% vs. 71.01%)”

[line 46-48; page 3]

The following quotation from the results section of this article reflects a more accurate treatment outcome in this population: “among estimated prevalence urban areas have a 24.84% achieving these targets compared with 25.16% in rural areas”.

[line 25-27; page 13]

The results section of the abstract has been revised to remove the (potentially misleading) existing statement, and this is replaced with the statement recommended by the reviewer (page 3):

“Results

Diabetes prevalence was higher in urban areas in England (7.80% [95% CI 7.30%; 8.31%]) relative to rural areas (7.24% [95% CI 6.32%; 8.16%]). For practices in urban areas, relatively fewer cases of diabetes were recorded in administrative medical records (69.55% vs. 71.86%), and a smaller percentage of those registered received the appropriate care (45.85% vs. 49.32%). Among estimated prevalent cases of diabetes, urban areas have a 24.84% achieving these targets compared with 25.16% in rural areas.”

Conclusion:

The following comment is potentially misleading: “Performance is considerably better than halving in terms of diagnoses in care records and achievement of treatment targets.”

[line 3-6; page 4]

The rule of halves is “Approximately half of most common chronic disorders are undetected, that half of those detected are not treated, and that half of those treated are not controlled”.

[Reference: Hart, JT. Rule of halves: implications of increasing diagnosis and reducing dropout for future workload and prescribing costs in primary care. British Journal of General Practice. 1992; 42:116-119]

Better than half of the estimated urban and rural diabetes population (69.55% vs 71.86%) are registered with clinical practices and have access to treatment. However, less than half of those registered for treatment have achieved treatment targets (35.71% vs 35.02%). [Supplementary table S2; page 26]

The adherence to/ or effectiveness of the local NICE guidelines is not a component of the rule of halves and is not the objective of the study. It is important to document that appropriate quality care/ adherence to these guidelines, when delivered to urban and rural patients (45.85% vs 49.32%), is associated with a greater likelihood of achieving treatment targets. (77.88% vs 71.01%).

We have now revised the conclusion section of the abstract to reflect the above points (page 4):

“The ‘rule of halves’ is not an accurate description of the actual pattern across the diabetes care pathway in England. More than half of the estimated urban and rural diabetes population are registered with clinical practices and have access to treatment. However, less than half of those registered for treatment have achieved treatment targets. Overall, performance across the levels of the rule of halves is lower for urban areas - but differences are explained by population characteristics and supply factors. Appropriate care quality was associated with a greater proportion of diabetic patients achieving treatment targets”

We have also now added in the quote from Hart’s paper as we thought this was an important point raised by the reviewer in that it would benefit the paper to clearly define the ROH for the reader in the introductory text (page 5):

“In practice, the ROH represents a broad ‘rule of thumb’ (8): “approximately half of most common chronic disorders are undetected, that half of those detected are not treated, and that half of those treated are not controlled”.”

4. Methods

The strategy for management of missing patient clinical data is not explicitly stated. Is this the definition of not achieving appropriate care?

That is the correct interpretation of the NDA indicators, and we have now stated this explicitly in the methods (page 9):

“Data on care quality from the NDA assume that patients without data for a measurement have not met the criteria for achieving an individual indicator (24).”

5. Ethics

No reference is made to ethical considerations.

This is now made clearer in the following statement (page 11):

“This study uses retrospective, observational data (most of which are at an aggregated level) and did not require ethical approval.”

11. Discussion

With regard to your comments: “Differences in the proportion of patients with a record of diabetes receiving the appropriate care could in theory reflect differences in attitudes and health literacy amongst urban and rural populations” [line 3-7 page 17]; and “This evidence could then be used to help inform development of evidence based health policy and clinical practice for urban populations for example by improving health literacy.” [lines 49-51; page 18]:

Do you have access to data on patient diabetes health literacy, e.g. participation in the NICE DESMOND program (diabetes education for self-management for ongoing and newly diagnosed)?

Unfortunately, we do not have timely access to these data, however, we have added a sentence later in the discussion noting that this could be a useful avenue for future studies (page 19):

“Future studies could also explore whether diabetes patients’ health literacy does in fact explain some of the variation between patients residing in urban and rural settings.”

Regarding: “Adjustment for the population characteristics of practices in urban areas substantially reduced the performance gap but did not fully account for the difference” [line13-18; page 17]: Information available to the investigators included the mental health status of urban and rural populations. [line 4-6; page 7] Was this included in further analyses?

There are data on mental health in the UKHLS. However, we cannot include this in the analyses as we would only be able to link mean levels of mental health variables in urban / rural residing respondents to the UKHLS to urban / rural practices from the NDA. These would be collinear with the binary indicator for urban/rural status in the regression analyses, and so cannot be included.

For reasons stated previously, consider amending this misleading comment in the conclusion: Performance is considerably better than having in terms of diagnosis in key records and achievement of treatment targets.
[line 37-39; page 18]

We have replaced the above with the following statement (page 19):

“More than half of the estimated urban and rural diabetes population are registered with clinical practices and have access to treatment. However, less than half of those registered for treatment have achieved treatment targets”

13. Funding

The source of funding is not defined.
[line 15-16; page 19]

This work was not directly funded. Rather, it was a ‘spin-off’ from work supported by the NIHR ARC for GM. We have captured this in the following statement (page 1 & 2):

“This research was supported by the National Institute for Health Research Applied Research Collaboration Greater Manchester. The views expressed in this publication are those of the author(s) and not necessarily those of the National Institute for Health Research or the Department of Health and Social Care. This research originates from work undertaken in the Cities Changing Diabetes (CCD) programme for Greater Manchester in partnership with and supported by Health Innovation

Manchester and Novo Nordisk. CCD was launched in 2014 by the Steno Diabetes Center Copenhagen, University College London, and Novo Nordisk.”

Reviewer: 3

Dr. Sheyu Li, Department of Endocrinology and Metabolism, West China Hospital

Comments to the Author:

The study based on a series of surveys illustrates the quality of care for diabetes in England, UK, and indicates some but not strong inequity of health care. There are some concerns regarding the analysis.

1. The quality of care is confusing and Table 1 needs citation to support the rationales. To be noted, ideal care strategy changes over time. The citation should fit the time point of care, and the time lag may be considered for the knowledge dissemination. To be noted, NICE is updating itself continuously at different aspects of diabetes care.

We have updated Table 1 to include the following footnote to include the relevant citations for the underlying indicators and data, and to clarify the guidelines are continually updated and how some indicators have changed since the time point (page 8):

“*notes: see (23) and (24). NICE recommendations have been revised since 2016-17 to recommend blood pressure targets at 140/90; and guidelines for best practice/care quality is continuously updated”

We have also added in a statement in the limitations section of the discussion to note the time lag of the data used in this study (page 19):

“Data on diabetes registrations, care processes and treatment targets were for the financial year 2016/17. However, the guidelines around best practice care are updated continuously and the findings of this study should be interpreted on this basis. ”

2. The data resource and definition for diabetes registration is critical and needs more clarification.

We have set this out more clearly (including the relevant citation) on pages 23 & 24:

“The 2016-17 NDA covers the majority of England (and Wales) with a participation rate of 95.3% (24). Information is collated from GP clinical systems and comparable data are collected from secondary care providers over a six week period. Data on registrations represent counts of patients who have a recorded diagnosis of diabetes in these data returns.”

3. The study differentiates only urban and rural areas in England. It might be more helpful to indicate the healthcare inequity across counties or shires of England.

We were unable to analyse counties and shires specifically due to the different administrative geographies at which the various data sources were available. However, we have increased the information provided on the patterns across different settings by repeating all descriptive and regression analyses (including corresponding versions of all tables and figures) now including the N=38 ‘urban with significant rural’ CCGs.

This material is presented in the results section of the supplementary material under the heading:

“Extension to include ‘urban with significant rural’ CCGs” (pages 24 to 34), and includes the following tables and figures:

- Supplementary Tables S4 - S8
- Supplementary Figure S1

This supplement is also referred to in the text in the data section which refers to the exclusion of the 38 ‘urban with significant rural’ CCGs (page 9):

“We excluded (N=38) CCGs classed as “urban with significant rural” to allow for comparison between more distinctly urban and rural CCGs (however, these CCGs are included in supplementary analyses).”

4. I am confused about the meaningfulness of the "ROH", especially the term "half". The paper says little about half, but different numbers for different events. I do not think that removing the term "ROH" eliminates the significance of the study.

We have added in additional clarity in the introductory text around how the ‘rule of halves’ is defined (page 5):

“In practice, the ROH represents a broad ‘rule of thumb’ (8): “approximately half of most common chronic disorders are undetected, that half of those detected are not treated, and that half of those treated are not controlled”.”

VERSION 2 – REVIEW

REVIEWER	Bernabe-Ortiz, Antonio Universidad Peruana Cayetano Heredia, CRONICAS Centre of Excellence in Chronic Diseases
REVIEW RETURNED	26-Nov-2021

GENERAL COMMENTS	Please delete the term "=1" for some variables (categories) in Tables.
--

REVIEWER	Dickson, Lynnsay University of the Witwatersrand, Paediatrics
REVIEW RETURNED	08-Dec-2021

GENERAL COMMENTS	You have presented clinically significant evidence that healthcare workers are not implementing established treatment protocols. This non-adherence/non-compliance to effective treatment protocols has the potential to cause harm to patients as well as increase health care costs. Consider including in ‘Implications for research and policy, further investigation of strategies to improve healthcare professional performance. The relatively punitive but effective strategy employed by Kaiser Permanente may not be universally applicable or feasible. (1) Reference Lester H., Fireman H., Campbell S., et. al. The impact of removing financial incentives from clinical quality indicators: a longitudinal analysis of four Kaiser Permanente indicators. BMJ 2010;340:c1898. doi:10.1136/BMJ.c1898
--

REVIEWER	Li, Sheyu Department of Endocrinology and Metabolism, West China Hospital, Sichuan University
REVIEW RETURNED	02-Dec-2021

GENERAL COMMENTS	Thanks for the active response from the authors, and happy to find the health equity between the rural and urban areas in England, UK. One minor issue: are the authors sure to start a sentence with "N=" in the abstract?
--